# 3D-Q-FISH/Telomere/TRF2 Nanotechnology Identifies a Progressively Disturbed Telomere/Shelterin/Lamin AC Complex as the Common Pathogenic, Molecular/Spatial Denominator of Classical Hodgkin Lymphoma

**DOI:** 10.3390/cells13211748

**Published:** 2024-10-23

**Authors:** Hans Knecht, Tina Petrogiannis-Haliotis, Sherif Louis, Sabine Mai

**Affiliations:** 1Division of Hematology, Department of Medicine, Jewish General Hospital, McGill University, Montreal, QC H3T 1E2, Canada; 2Department of Pathology, Jewish General Hospital, McGill University, Montreal, QC H3T 1E2, Canada; tina.haliotis@mcgill.ca; 3Telo Genomics Corp., Ontario, ON M5G 1L7, Canada; sherif.louis@telodx.com; 4Department of Physiology and Pathophysiology, University of Manitoba, Winnipeg, MB R3T 2N2, Canada; sabine.mai@umanitoba.ca

**Keywords:** Hodgkin lymphoma, Reed–Sternberg cell, nanotechnology, 3D Q-FISH, telomere/shelterin, TRF2, lamin A/C, LMP1 oncogene, breakage–bridge–fusion (BBF) cycles, t-stumps

## Abstract

The bi- or multinucleated Reed–Sternberg cell (RS) is the diagnostic cornerstone of Epstein–Barr Virus (EBV)-positive and EBV-negative classical Hodgkin lymphoma (cHL). cHL is a germinal center (GC)-derived B-cell disease. Hodgkin cells (H) are the mononuclear precursors of RS. An experimental model has to fulfill three conditions to qualify as common pathogenic denominator: (i) to be of GC-derived B-cell origin, (ii) to be EBV-negative to avoid EBV latency III expression and (iii) to support permanent EBV-encoded oncogenic latent membrane protein (LMP1) expression upon induction. These conditions are unified in the EBV-, diffuse large B-Cell lymphoma (DLBCL) cell line BJAB-tTA-LMP1. 3D reconstructive nanotechnology revealed spatial, quantitative and qualitative disturbance of telomere/shelterin interactions in mononuclear H-like cells, with further progression during transition to RS-like cells, including progressive complexity of the karyotype with every mitotic cycle, due to BBF (breakage/bridge/fusion) events. The findings of this model were confirmed in diagnostic patient samples and correlate with clinical outcomes. Moreover, in vitro, significant disturbance of the lamin AC/telomere interaction progressively occurred. In summary, our research over the past three decades identified cHL as the first lymphoid malignancy driven by a disturbed telomere/shelterin/lamin AC interaction, generating the diagnostic RS. Our findings may act as trailblazer for tailored therapies in refractory cHL.

## 1. Introduction

Hodgkin lymphoma (HL) was first described in 1832 [1], and its diagnostic tumor cell, the bi- or multinuclear Reed–Sternberg (RS) cell, was discovered independently by an Austrian [2] and an American [3] pathologist in 1898 and 1902, respectively. The precursor of the RS-cell is the mononuclear Hodgkin (H) cell, itself originating from germinal center (GC) B-lymphocytes [4,5,6,7]. As a particularity, circulating small clonotypic precursor B-cells are very rare or even undetectable [8]. In about 40–50% of HL cases, H- and RS-cells express the Epstein–Barr virus (EBV)-encoded latent membrane protein 1 (LMP1) or its deletion variants [9,10]. The WHO Classification of Tumors of Haematopoietic and Lymphoid Tissues [11], divides HL into nodular lymphocyte-predominant Hodgkin lymphoma (NLPHL) and classic Hodgkin lymphoma (cHL). Histologically, most cHL cases are characterized by a low number of H-cells and even less frequent bi- or multinuclear RS-cells in a mixed background of non-neoplastic T-lymphocytes and inflammatory and accessory cells. cHL is subdivided into four histological subtypes: nodular sclerosis (NS), mixed cellularity (MC), lymphocyte-rich (LR) and lymphocyte-depleted (LD). Regardless of subtype, H- and RS-cells express an immune-profile that is typically CD30+, CD15+, CD45 LCA-, CD20-, PAX5+ (weaker than in nonneoplastic B-cells), MUM-1+, OCT-2- and BOB-1-. In EBV-associated cHL, H- and RS-cells are typically EBER+ and LMP1+ (Figure 1), and the postulated lymphoid precursor is a pre-apoptotic GC B-lymphocyte, rescued by EBV-encoded latent genes [6,7,11].

Though nearly all (98–99%) cHL cases are of the B-cell genotype, very rare cases of cHL, maximally 1–2%, harbor a true T-cell genotype in their H- and RS-cells [13,14,15], pointing to a common pathogenic denominator, basically independent of a B- or T-cell genotype. The hypothesis of a basic molecular mechanism as a common pathogenic denominator is also supported by the fact that transfection of the LMP1 oncogene in the EBV- Hodgkin cell line L-428 and the mononuclear human embryonic kidney cell line 293 induces RS and RS-like multinucleated cell formation, respectively [16]. H- and RS-cells present numerical chromosome aberrations, are in the hyperploid range [15,17], and exhibit a complex and unstable karyotype [18,19]. However, despite recurrent translocations and gene amplifications [20,21], a cHL-defining translocation has not been identified [22].

Thanks to a myriad of epidemiological, molecular and genetic approaches, our understanding of the specific nature of cHL has considerably advanced in the past three decades, and several comprehensive reviews document this progress [23,24,25,26]. Already in the year 2000 we postulated the identification of a common pathogenic denominator, closely connected to targets of the LMP1 oncogene [27], based on the oncoprotein’s permanent activation of the NF-kappa-B, JAK3-STAT and c-Jun signaling pathways, which are mandatory for H and RS formation [5,22,25,26]. The NF-kappa-B, JAK3-STAT, PI3K/AKT and c-Jun signaling pathways are also permanently up-regulated in EBV-negative cHL [24,25]. Dysregulation of the STAT, ATM, FOX and P53 pathways has also been described irrespective of the EBV status [22,25]. Chromosomal breakpoints affecting the Ig loci consistent with class switch recombination defects are present in nearly 20% of cHL cases [20]. Alterations of the PD-L1 and PD-L2 expression in H- and RS-cells are found in over 90% of cHL cases, and chromosome 9p24.1 amplification is associated with significantly shorter survival and advanced stage disease [21]. Because these changes are independent of the EBV status of H- and RS-cells, but LMP1 expression also dysregulates most of these essential signaling pathways, we hypothesized that LMP1 also alters a fundamental pathway in cellular biology, the dysregulation of which in the long run might be involved in the high number of genetic and signaling abnormalities known in EBV+ and EBV- cHL. However, the identification of this common pathogenic denominator at the origin of the H- and diagnostic RS-cells was only recently achieved. The development of a highly accurate 3D nanotechnology approach, applicable to cytospins and deparaffinized histologic diagnostic slides [28,29,30,31], and the molecular and structural characterization of the telomerase/shelterin complex [32,33,34,35] allowed this progress.

Telomeric DNA and the shelterin complex consist of multiple (TTAGGG)n repeats ending in a single-stranded overhang of the G-rich 3′ strand, and a number of specific proteins called shelterins, either binding telomeric DNA directly or being associated with telomeric chromatin, are found on telomeres [32]. The six human shelterin proteins are TRF1 (telomeric repeat binding factor 1), TRF2 (telomeric repeat binding factor 2), POT1 (protection of telomeres 1), TIN2 (TRF1 interacting nuclear protein 2), TPP1 (TIN2 interacting protein 1) and RAP1 (repressor activator protein 1). TRF1, TRF2 and POT1 directly interact with telomeric DNA, whereas TIN2, TPP1 and RAP1 interact sterically with the former three to form the 3D telomeric complex [33]. Amongst the shelterin proteins, TRF2 has emerged as a key player; in particular, TRF2 is both necessary and sufficient for t-loop configuration, and it protects this structure. TRF2 suppresses ATM (ataxia-telangiectasia-mutated) signaling and therefore classical nonhomologous end joining (c-NHEJ), but ATM is activated at telomeres lacking TRF2 [33]. TRF2 has also a central role in interaction with nucleosomes, the basic element of chromatin organization, and telomere/shelterin complexes are dynamic structures [34].

## 2. Review

### 2.1. LMP1: The Golden Key to H and RS in 2D-Restricted Research

The nucleotide sequence and protein structure of LMP1, encoded by the BNLF1 gene of EBV, were discovered in 1984 [36]. A year later, LMP1 was identified as a viral oncogene by its capacity to transform rodent fibroblasts and to render them tumorigenic in nude mice [37]. Experimental LMP1 expression in primary human GC B-cells induces transcriptional profiles characteristic of Hodgkin cell lines, in particular the loss of B-cell antigens such as CD19, CD20 and CD79 but also the induction of ID2-hampering B-cell differentiation [38]. In mononucleosis infectiosa (IM), type II latency B-lymphoid cells (EBNA2-, LMP1+) are regularly observed [39], and lymph node biopsies of IM contain few binucleated LMP1+ cells with characteristic RS owl-eye cytomorphology [40]. Most importantly, the naturally occurring EBNA2-deleted EBV strain P3HR1 causes, in a subset of infected cord-blood humanized mice, Hodgkin-like type II latency B-cell lymphomas with numerous bi- and multinucleated LMP1+, CD30+, CD45- and RS-like cells [41]. From these findings it appears evident that permanent LMP1 expression is an inducer of multinuclearity and the characteristic cytomorphology of RS. But which steps finally led to genomic instability, such as a numerical increase in chromosomes, increasingly complex cytogenetics and multinucleation, had yet to be determined.

The LMP1-induced deregulation of key components of the shelterin complex, the safeguard of telomeres [32], appears as a major element in this puzzle [42]. Lymphoblastoid cell lines (LCL) from EBV- healthy donors, infected with EBV strain 95.8, develop after 4 weeks deletions, fragments of chromosomes, dicentric chromosomes, unbalanced translocations and chromatid gaps when karyotyped by SKY (spectral karyotyping) [42]. These structural chromosomal anomalies are not identifiable in normal, mitogen-stimulated EBV- lymphocytes [42]. In LCL numerous foci of γH2AX and of MRE11, indicating DSBs (double-strand breaks) and DNA damage responses, respectively, occur but are not found in the mitogen-stimulated control lymphocytes. Moreover, about 75% of the analyzed LCL nuclei contain TRF2 (telomere repeat binding factor 2)-free telomeres, which is consistent with telomere uncapping. Based on these seminal 2D papers [16,27,32,41,42], it is evident that the target genes of the multifunctional LMP1 oncoprotein are intrinsically associated with the generation of H- and RS-cells, as suspected already two decades ago [27,43]. Uncovering the mechanisms of these interactions/steps was only made possible through the progressive development of 3D telomere q-FISH [28,29,44,45] over the last decade.

### 2.2. 3D Telomere-Shelterin Complex: The Railway Turntable in H and RS Morphogenesis

Small-telomere and telomere-poor Hodgkin cells were first identified by 2D telomere-PNA-FISH by Balta-Yildirim in cHL lymph node touch preparations and HDLM-2 cytospins [46]. Major progress was only achieved through 3D nanotechnology analysis of the nuclear spatial distribution and size of the telomeres and telomere aggregates (clusters of telomeres found in close association which are not identifiable as separate entities due to the optical resolution limit of 200 nm [47]) combined with immunohistochemistry in three H-cell lines (HDLM-2, L-428, l-1236) and three diagnostic lymph node biopsies (two NS, one MC) [48]. This work uncovered the mechanistic steps of the transition from H- to RS-cells [48]. RS-cells of the three H-cell lines show significantly shorter and significantly fewer telomeres in relation to the total nuclear volume when compared to H-cells [48]. (Cytospins allow analysis of the entire volume of H- and RS-cells by a 3D technique, whereas 3D analysis of 5 μ of histologic sections is limited to a segment of the nucleus/nuclei, given the nuclear diameter of H- and RS-cells is much greater than 5 μ). In RS-cells of HDLM-2, multiple pairs of centromeres (anti-centrin-2 antibody) and multiple atypical spindles (anti-γ-tubulin) are identified. γH2AX immune staining, indicating double-strand DNA breaks, is most impressive in giant RS nuclei but only faintly present in H nuclei. Multiple atypical spindles are also present in large RS-cells of the diagnostic biopsy of the MC case. These centrosome and mitotic spindle abnormalities confirm our earlier 2D observations in L-428 cells [49] but add the 3D configuration, number and size of telomeres as new essential elements in the transition of H to RS. In all three H-cell lines, but also in the diagnostic biopsies of one NS case and the MC case, the transition from H to RS was characterized by a significant increase of short and very short telomeres, so-called t-stumps [50] and the emergence of virtually telomere-free “ghost” nuclei in RS [48]. Analogous findings were observed in a fourth H-cell line, U-HO1, from primary refractory Hodgkin lymphoma [51]. The 3D kinetics were partially reversible in U-HO1-PTPN1 (non-receptor-protein-tyrosine phosphatase N1), a stably transfected daughter H-cell line of U-HO1. Culture of U-HO1-PTPN1 cells induced de-phosphorylation of STAT5 with consecutive lack of Akt/PKB activation and cellular arrest in G2, promoting induction of apoptosis. Contrary to U-HO1 with its high STAT5A expression, U-HO1-PTPN1 was characterized by nearly absent STAT5A expression, three times longer doubling time, accumulation of RS-cells, prevention of “t-stump” formation and fourfold increases of apoptotic H- and RS-cells. These findings confirm that STAT5 is essential for cHL [52] but also open the transition from H to RS as a target for new therapeutic research.

When analyzing 3 LMP1+ and 3 EBV- cHL cases, the primary 3D findings in HDLM-2, L-428, L-1236 and U-HO1 were further confirmed [53]. Our 3D telomere FISH findings on H-cell lines and diagnostic lymph node biopsies define RS-cells as end-stage tumor cells because their telomere loss, increase of “t-stumps”, aggregate formation and generation of “ghost” nuclei will no longer sustain chromosome segregation and the ability to divide further [48]. In our personal experience, this limit for RS in diagnostic biopsies is up to 10–12 nuclei [10].

### 2.3. 3D Nuclear Remodeling During the Transition from H to RS

Maintenance of proper nuclear architecture is essential for proper cell function including mitosis, transcription, translation and genomic stability [54,55,56]. Chromosomes are organized in a non-random manner to assess correct cellular function, as convincingly shown by the different architecture of rod cell nuclei in animals with diurnal and nocturnal vision [57]. Significant 3D nuclear remodeling is also identified during the transition from H to RS [58]. Using a fixation protocol maintaining the 3D interphase nuclear structure [59], we analyzed the positions of chromosomes 9 and 22 in the H-cell lines HDLM-2, L-428 and L-1236 [58], using healthy lymphocytes (neighboring chromosomal territories) and CML cells (overlapping due to the t(9;22)) as negative and positive controls, respectively. Significant differences in the chromosome territories were observed in H-cells for both chromosomes and were progressing when transforming to RS (for example, in a three-nucleated RS-cell, most chromosome 9 and 22 content was present in one nucleus, with the two others nearly free of 9 and 22 content, consistent with “ghost” nuclei [58]). Chromosome painting of metaphase spreads in HDLM-2 identified the presence of chromosomes with a “zebra” stripe-like pattern, typical for chromosomes having undergone several rounds of BBF. Spectral karyotyping (SKY) confirmed “zebra” chromosomes involving different chromosome partners in L-1236 [58]. The complexity of chromosomal rearrangements progresses from H to RS; for example a t(1;17) found in H progresses to a t(1;17;13;18) or t(1;17;18;19).

Ongoing nuclear remodeling was confirmed by 3D SIM (structured illumination microscopy). Nuclei of RS-cells were frequently linked to each other through internuclear bridges, consisting of stretched DNA fibers and/or individual chromosomes. DNA free spaces “holes” increased from H to RS and progressed further from bi- to tetranuclear RS [60]. The disruption of nuclear architecture when progressing from H to RS, as suggested by our group earlier [48], was confirmed. Moreover, we documented complex chromosome dynamics during the transition from H to RS [58].

### 2.4. LMP1 Induces TRF2 Deregulation as Essential Step for H Formation and Progression to RS

TRF2 was identified in 1992 independently by a French [61] and an American [62] group. The main functions of TRF2 are the prevention of DSB repair activities, protection of eukaryotic chromosome ends and nuclear envelope interactions [63,64,65]. Based on our previous clinical experience, immune-histologic 2D findings and 3D nanotechnology results, it was highly probable that the LMP1 oncoprotein targeted directly or indirectly key proteins of the shelterin complex and, upon reactivation in GC-derived B-cells, induced H and RS through permanent activation. To test this hypothesis, we needed a long-term tet-off inducible LMP1 expression system in a GC-derived B-cell line. The BJAB-tTA-LMP1 cell line and its negative control BJAB-tTA correspond to these prerequisites [66].

Indeed, permanent expression of LMP1 in the BJAB-tTA-LMP1 cell line induces significant down-regulation of telomere repeat binding factor 1 (TRF1), TRF2 and protection of telomere 1 (POT1) at both the transcriptional and translational level (the most impressive down-regulation (>60%) was observed in TRF2), resulting in a highly significant increase of multinucleated, LMP1+ RS-like cells (*p* < 0.0001) on days 14 and 21 [67]. Moreover, this LMP1-induced significant (*p* < 0.05) down-regulation of TRF1, TRF2 and POT1 at the transcriptional and translational level was reversible upon re-suppression of LMP1 (tet-on) at day 3, according to measurements on days 7 and 14. The same was true for suppression on day 7, when measured on day 14 [67].

Because conditional TRF2 deletion elicits an ATM-mediated telomere damage response, with γH2AX up-regulation resulting in telomere fusions and giant chromosomes [68], we performed SKY on 20 metaphases each of BJAB-tTA-LMP1 expressed (tet-off) on day 1 and on day 20, and on 20 metaphases of BJAB-tTA-LMP1 suppressed (tet-on) on day 20. The variation in the chromosome number expressed in LMP1 was small at day 1 (44–58 chromosomes) but major at day 14 (19–316 chromosomes), indicating the presence of “ghost” and giant RS-like cells. The number of “zebra” chromosomes increased by a factor of three, indicating ongoing BBF cycles [67]. Thus, LMP1 induced down-regulation of TRF2 results in complex cytogenetics, a hallmark of most RS-cells.

To test the hypothesis that TRF2 was intrinsically associated with the generation of multinucleated RS-like cells, we established a stable BJAB-tTA-LMP1/mycTRF2 transfectant capable of producing myc-driven TRF2 independent of the LMP1-mediated suppression. And indeed, the myc-driven TRF2 compensated for the TRF2 suppression induced by LMP1 (tet-off). The formation of multinucleated RS-like cells was blocked, proving that TRF2 was essential to prevent multinucleation [67]. Again, LMP1-induced multinucleated RS-like cells showed uneven telomere distribution, nearly absent TRF2 at day 14 and abundant “t-stumps” at day 21 when compared to LMP1-suppressed cells. Thus, the generation of unprotected telomeres was induced by LMP1. Analogous findings were confirmed by combined 3D telomere FISH-TRF2 immunohistochemistry in primary H- and RS-cells of EBV-cHL [67], underscoring a common pathogenic denominator, mostly consistent with the telomere–shelterin complex.

### 2.5. Disruption of Direct 3D Telomere–TRF2 Interaction Is a Hallmark of H and RS

In order to confirm our in vitro post GC B-cell model of EBV-associated cHL, where the LMP1 oncogene mediates multinuclearity through down-regulation of TRF2, we further developed and adapted our 3D combined quantitative TRF2-telomere immuno Q-FISH protocol (3D TRF2/Telo-Q-FISH) [29,45,47,67] to monolayers of H- and RS-cells including the surrounding lymphocytes [69]. Cytocentrifuged monolayers of lymph node suspensions from 14 diagnostic cHL biopsies (4 LMP1-expressing) were analyzed. The reactive surrounding lymphocytes as well as benign BJ-5ta fibroblasts served as controls. This approach allowed for the first time the 3D analysis of the entire nuclear content of H- and RS-cells [69]. Of note, the entire nuclear analysis is not achieved with laser dissection, performed on 5 µm sections, given that the nuclear diameter of H- and RS-cells generally exceeds 10 µm.

As expected, in all LMP1+ and EBV- cases, the internal control lymphocytes and the BJ-5ta fibroblasts showed a tight 1:1 telomere–TRF2 association, excluding technical pitfalls in the H and RS analysis. On the contrary, all H and RS (193 H and 122 RS) of all 14 cHL cases showed an unambiguous disruption of the direct, quantitative and qualitative telomere/TRF2 interaction [69]. Interestingly, two, at first glance opposite-appearing patterns were observed. The four LMP1-expressing and four EBV- cases (total eight cases) were characterized by a most significant attrition of TRF2 spots during the transition from H to RS (for example, a ratio of telomere/TRF2 signals from 2.5 in H to 3.8 in RS), leaving most telomeres de-protected in RS-cells, designated as Pattern B. In Pattern B, “ghost” RS-cells with virtually absent TRF2 and telomere signals were regularly observed, but “ghost” H-cells with very large nuclei (15–24 µm) were rare (Figure 2).

Six cases (all EBV negative) were characterized by significantly more free TRF2 spots than telomere signals, identified as Pattern A. Some signals co-localized, but there were always also free telomere signals, consistent with de-protection [69]. Again, there was progression of the signal ratio of telomere/TRF2 with the transition from H to RS (for example, 0.5 in H to 0.2 in RS). Most of the remaining telomere spots were “t-stumps”. Nuclei of RS with Pattern A had often intranuclear DNA bridges. Interestingly, in vitro overexpression of TRF2 leads to replication stalling, chromosome end-to-end fusions and loss of telomeric sequences [70]. Ultrafine anaphase bridges with loss of telomeric sequences and telomere shortening occur rapidly. Significantly elevated TRF2 levels also mechanistically induce genomic instability [70], and very short telomeres are fused by both the classical and alternative NHEJ (non-homologous end joining) pathways [71]. To the best of our knowledge, the reason for this high TRF2 expression in H- and RS-cells of some EBV- cHL cases is not known. Thus, in addition to TRF2 down-regulation, TRF2 overexpression also leads to genomic instability, the driving force in refractory cHL. The defective 3D steric interaction of telomeres with shelterin proteins, especially TRF2, an essential element in the molecular pathology of cHL, was only recently recognized by other research groups [23,26,41]. Figure 3 summarizes the main events of the molecular, mechanistic pathogenesis of EBV-positive and some EBV-negative cHL, based on our 3D experimental and translational findings.

### 2.6. Lamin Basics and Lamin A/C Overexpression in H and RS

Lamins are type V intermediate filament proteins, associate physically with the inner nuclear membrane, tether heterochromatin in the periphery and exert important scaffolding functions within the nucleus [76,77,78]. In human cells, two types of lamin have been identified: B-type lamins, encoded by the *LMNB1* gene for lamin B1 [79] and the *LMNB2* gene for lamin B2 [80], and A-type lamins, encoded by the *LMNA* gene, the alternative splicing of which produces lamin A and lamin C [81]. Recently 3D-SIM has allowed the characterization of a distinct fiber meshwork at the supramolecular level for lamins A, C, B1 and B2 [82], and cryo-electron microscopic tomography has identified a coil-coiled, 51 nm long and 3.5 nm thick rod-like dimer with two lateral globular domains as the building blocks of all lamins [83,84]. Lamin dimers form head-to-tail polymers, which laterally interact to form proto-filaments [84,85]. Stochastic optical reconstruction microscopy (STORM) further revealed that lamin B1 and lamin A/C form two concentric and overlapping nutshell-like networks adjacent to the inner nuclear membrane (INM) [86]. Lamin B1 preferentially localizes directly adjacent to the INM, whereas lamin A/C localizes closer to the nucleoplasm [86]. Lamin B1 and lamin B2 are constitutively expressed and appear to be necessary for cell survival [87], whereas lamin A/C, also distributed throughout the nuclear interior [77], is involved in transcription [88] and plays a crucial role in the regulation of mitotic spindle assembly and positioning [89]. Distinct mutations in the coding sequence of the latter result in severe congenital disorders known as laminopathies [90], with the most impressive being Hutchinson–Gilford progeria syndrome (HGPS) [91].

Already in 1997, Jansen and coworkers [92] documented through immunohistochemistry strong lamin A/C expression in H and RS but no expression in centrocytes, centroblasts and mantle zone lymphocytes of reactive lymph node follicles. Lamin A/C was also identified in stimulated blood lymphocytes [93,94], and its overexpression was confirmed by 3D nanotechnology in H- and RS-cells by our group [94]. The regular, spherically shaped lamin A/C organization, identified in activated lymphocytes, is disturbed in mononuclear H-cells and undergoes further structural changes during the transformation of H to binuclear and finally multinuclear RS [94]. This dynamic process of steadily increasing lamin A/C-associated nuclear compartmentalization of individual nuclei was identified in both H-cell lines (Appendix A in form of HDLM-2 Video Clip A H-cell, and Video Clip B RS-cell) and H and RS of diagnostic lymph node biopsies [31,94].

### 2.7. Progressive Disruption of the Lamin A/C–TRF2–DNA Interaction from H to RS

There is increasing evidence of tight interactions of lamins with telomeres and the shelterin complex. Human fibroblasts with a homozygous nonsense mutation of the lamin A/C gene, and thus completely devoid of lamin A/C expression, show increased nuclear plasticity and telomere mobility [95]. Chromatin immuno-precipitation (ChIP) assays performed with lamin A/C antibody revealed binding of lamin A/C to telomeres, and expression of a dominant negative TRF2 mutant leads to loss of telomere integrity and to the formation of DNA damage foci at telomeres [96]. Indeed, TRF2 is not only essential for the formation and maintenance of the configuration of functional telomeres at chromosomal ends [97], but it also binds to interstitial telomeric sequences (ITS) [98]. The interaction of TRF2 with lamin A/C occurs at the long linker region of TRF2 [99]. On the other hand, lamin A/C is essential for maintaining the chromosomal nuclear territories, and this occurs partially through direct interaction with TRF2 bound to t-loops at ITS [100], so-called interstitial t-loops (ITL). Impairment or loss of this 3D lamin A/C–TRF2 interaction at ITL results in increased chromatin dynamics [101], changes in gene expression and impaired chromosome stability and genomic integrity [102,103].

Considering the experimental data on TRF2 down-regulation [64,65,67] and overexpression [70,71] in cellular 3D structures, we suspected the direct 3D interaction of lamin A/C with TRF2, localized at t-loops of telomeres and ITS, to undergo substantial changes during the formation of H and further transformation to RS. We applied our combined quantitative 3D TRF2–telomere immuno Q-FISH (3D TRF2/Telo-Q-FISH) protocol [45] and extended it to lamin A/C as a further target. Our findings revealed very high and abnormal lamin A/C protein expression and a completely disrupted lamin A/C–TRF2–ITL/telomere interaction in H- and RS-cells (abnormal internal lamin A/C structures within the nucleus) in contrast with BJ-5ta control fibroblasts (Figure 4) [24,31,104]. A significant increase of lamin A/C accumulation at the RNA level, and even more impressively, at the protein level, from normal lymphocytes (at 1) to H and finally RS (over 20) was observed [94,104]. This is consistent with progressive destruction of the nuclear architecture from H to bi- and finally multinuclear RS.

### 2.8. Prognostic and Future Potential Therapeutic Implications of the 3D Nanotechnology Findings

The internationally most commonly used chemotherapy regimen for cHL is ABVD (Adriamycin (doxorubicin), bleomycin, vinblastine, dacarbazine) [105,106,107,108]. In two large studies, disease-free survival at 4 years was 85.8% [109], and progression-free survival at 10 years was still high at 69% [110], not inferior to more recently developed chemotherapy regimens. However, despite this high success rate, a reliable upfront biomarker to identify primary refractory and early relapsing patients is still an unmet need. We are confident that cHL patients with a high percentage of t-stumps in the diagnostic biopsy [111], heralding high risk for early relapse or ABVD refractoriness [111], might see upfront benefits from brentuximab vedotin and/or recently developed checkpoint inhibitors like pembrolizumab and nivolumab [112,113,114] or more costly treatments as allo-hematopoietic stem cell transplantation and cellular therapies [115].

A recently published international retrospective study [116] confirms the findings from our pilot study [111]. In a cohort of 156 cHL patients with initial ABVD therapy, 126 patients were in complete remission for at least five years, and 30 patients were refractory or relapsing (RR) within 12 months. The percentage of t-stumps was by far the most prominent predictor in identifying RR cHL patients prior to the initiation of ABVD therapy, confirming the percentage of very short telomeres (t-stumps) as reliable biomarker. This appears to be convincing because the length of telomeres is related to cell division cycles [117,118]; in other words, the more divisions, the shorter the telomeres [119,120]. Thus, in aggressive cHL the mononuclear H-cells are characterized by a very high proliferation index, irrespective of the clinical disease stage. The model characteristics included an AUC of 0.83 in ROC analysis and a sensitivity and specificity of 0.82 and 0.78, respectively [116]. This is a predictive model which might be tested in a prospective study with newer chemotherapy and immune-chemotherapy protocols including checkpoint inhibitors.

The 3D telo-immuno-Q-FISH protocol [116] is easy to perform in a cytogenetic laboratory. It is a FISH-based assay with microscopy and automated analysis (TeloView^®^ software platform (v1.03), Telo Genomics, Toronto, ON, Canada). An established cytogenetics laboratory is capable of performing this assay routinely. We published a detailed methodology manuscript in Methods in Molecular Biology in 2017 [45]. In addition to the success in deciphering important facets of the molecular pathogenesis of H and RS [48,51,63,67,69,111,116], our Q-FISH protocol has been successfully applied in prostate carcinoma [121], myelodysplasia [122], cutaneous T-cell lymphoma [123] and very recently as biomarker in multiple myeloma [124]. 3D telomere profiling allows the prediction of the risk of progression from smoldering myeloma to frank multiple myeloma [124]. In cHL we are near to stratifying the risk to progress rapidly, when ABVD is used as first chemotherapy [111,116]. However, prior to establishing clear cut-off values (high versus low risk), we would need a further prospective international study with a large cohort treated upfront with ABVD. The 3D telo-immuno-Q-FISH protocol has without a doubt shown itself to be a powerful tool in elucidating pathogenetic mechanisms in various forms of cancer and is on the way to being accepted as a reliable biomarker. However, to achieve this goal, we need further prospective international cohort studies in more frequent forms of cancer like prostrate carcinoma and multiple myeloma, as well as rare diseases such as cHL.

fBALM (fluctuation-assisted binding-activated localization microscopy) may represent a game-changing technology in the study of nanomolecular structures and drug design for a wide variety of pathological conditions. This technique has been successfully applied to H and RS [125]. With a resolution of around 50 nm, the therapeutic effects of small selective molecules on nuclear sites of DNA–protein interactions could be visualized. Inhibition of telomerase using BIBR1532 followed by ALT (alternative lengthening of telomere) inhibition by trabectedin—the ALT pathway is activated in advanced cHL [126]—caused a decrease of greater than 90% in cell viability in three patient-derived HL cell lines [127]. Thus, the 3D shelterin/telomere nanotechnology approach and fBALM appear to be promising methods for future small selective molecule treatment evaluation [128].

## 3. Conclusions

In the present review we have focused on the 3D nanotechnology-based progress in the elucidation of the molecular steps leading from an activated B-lymphocyte to H-cell and finally diagnostic RS-cell formation in cHL. Our 3D analysis of the shelterin–telomere–lamin A/C complex over the past two decades was the trailblazer of this discovery. The “golden key” to this advancement was the LMP1-driven identification of TRF2 as the master protein in the transformation from H to RS. A new player in the field is lamin A/C interacting with TRF2. Our research also allowed the identification of telomere t-stump quantification as an upfront biomarker in cHL. We are confident that the sensitivity and specificity of this biomarker will increase in association with recently developed molecular biomarkers such as circulating tumor DNA quantification [129] and be useful for clinical applications. Moreover, in the near future, 3D analysis of recently discovered TRF2/telomere-interacting proteins [130] and columnar stacking of telomeric chromatin mediated by TRF2 [131] will foster progress.

## Figures and Tables

**Figure 1 cells-13-01748-f001:**
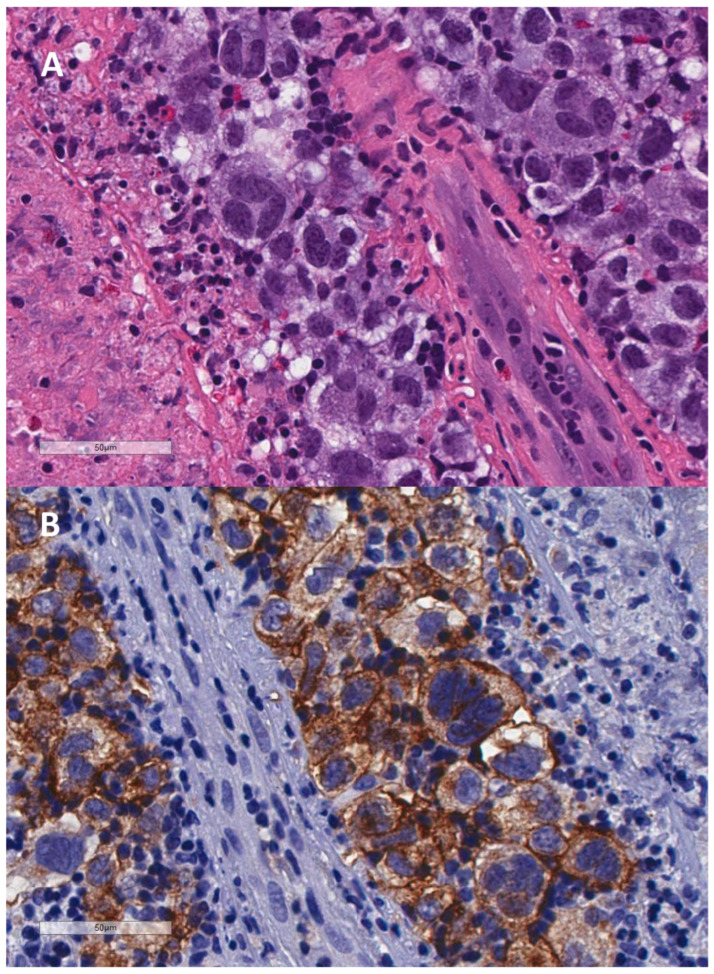
Very rare brain localization in a stage IVB case of EBV+ cHL, lymphocyte depleted, in an immune-compromised HIV+ patient. Note the nearly syncytial proliferation of H- and RS-cells, as described in a case report [12], and mainly absent reactive lymphocytes. Original magnification 40×. (**A**) Standard H&E staining. (**B**). Immune staining with anti-CD30 MoAb; (**C**) EBER in situ hybridization demonstrating the EBV positivity of H- and RS-cells but also some small lymphoid precursors. (**D**). Immune staining with anti-LMP1 MoAb. Note that LMP1 expression is only observed in H- and RS-cells, whereas small lymphoid precursors are still LMP1-. EBV+ H- and RS-cells in addition expressed CD15, MUM-1, PAX-5 and Ki-67.

**Figure 2 cells-13-01748-f002:**
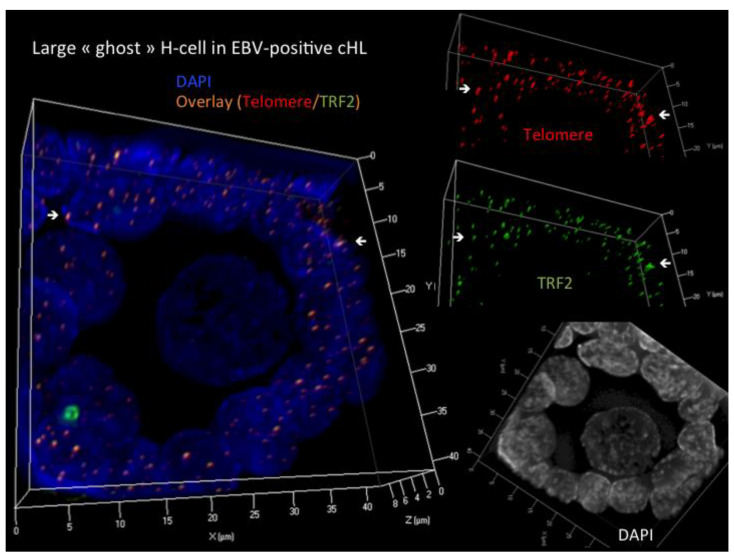
There is a complete loss of telomeres and TRF2 in a large LMP1-expressing H-cell, whereas the surrounding lymphocytes show a normal, tight 1:1 3D telomere/TRF2 interaction. White arrows identify corresponding (identical) telomere/TRF2 spots. Lower right is a DAPI staining in transparency mode to optimize cyto-morphological aspects. Upper right (telomere) and middle right (TRF2) demonstrate that the surrounding lymphocytes contain mainly small to mid-sized telomeres with a preserved 1:1 3D interaction with TRF2. The left panel (transparency mode) reveals a true “ghost” H-cell virtually without any TRF2 or telomere signals. The lymphocyte corona serves as an internal control.

**Figure 3 cells-13-01748-f003:**
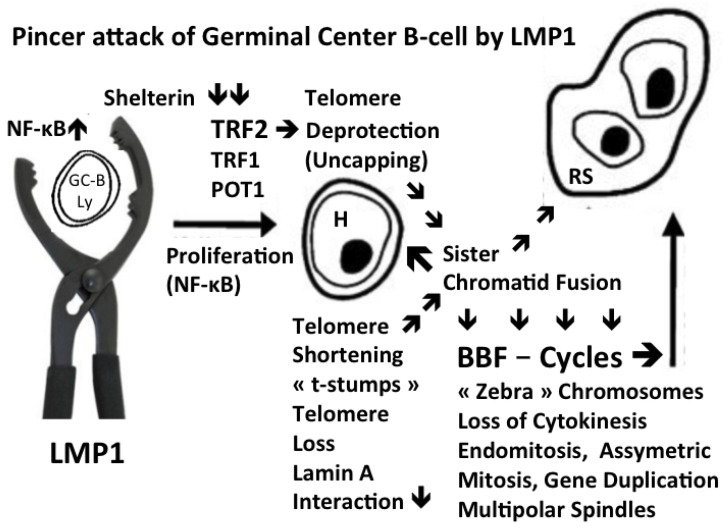
Shown from left to right: a continuous transformation from a GC B-lymphocyte into an H-cell and a terminal RS-cell. A “crippled” GCB-lymphocyte [6] or an EBV+ memory B-lymphocyte [7] re-entering GC reaction undergoes LMP1 oncoprotein-induced transformation, targeting the telomere/shelterin complex through a “pincer attack”. NF-kB-driven mitosis [72] combined with TRF2 down-regulation [48,67] leads to telomere shortening and de-protection, followed by sister chromatid fusion at the origin of repeated BBF cycles, resulting first in H-cells, and, after several subsequent mitotic cycles, in the end-stage multi-faceted, often telomere-poor RS-cells [46,48,53,58]. The constitutive NF-kB activation [72,73]: the dysfunctional telomere–shelterin complex, in particular TRF2 [67,69] appears to emerge as a logical candidate for a common pathogenic denominator not only for EBV-positive [74] but also for many TRF2-poor EBV-negative cHL cases [75].

**Figure 4 cells-13-01748-f004:**
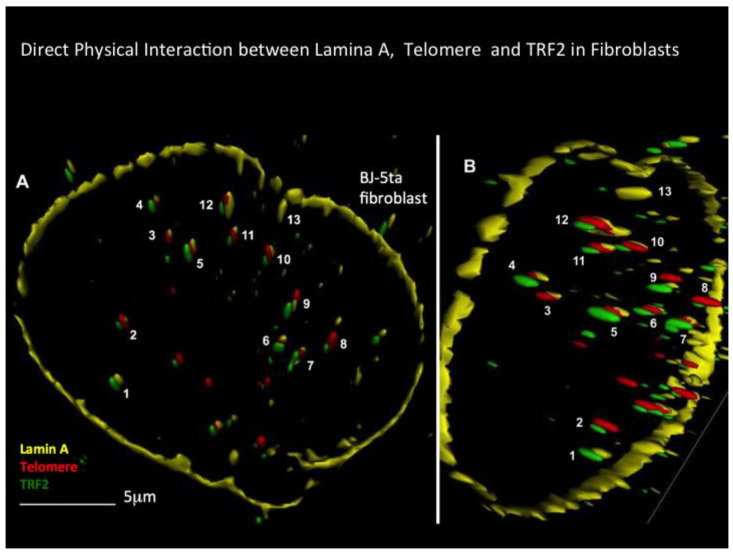
3D lamin A–telomere–TRF2 in surface mode reconstitution in a benign BJ-5ta fibroblast at different angles (**A**,**B**) is shown; 1–13 are corresponding spots. Direct 3D interaction of lamin A–telomere–TRF2 is seen at spots 2, 5, 6, 7, 10, 11 and 12. Direct lamin A–telomere interaction is seen at spots 3 and 4, and direct lamin A–TRF2 interaction is seen at spot 1. This type of interaction is completely disrupted in H and RS [24,104].

## Data Availability

Not applicable.

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
