# Peer review of "3D-Q-FISH/Telomere/TRF2 Nanotechnology Identifies a Progressively Disturbed Telomere/Shelterin/Lamin AC Complex as the Common Pathogenic, Molecular/Spatial Denominator of Classical Hodgkin Lymphoma"

_cells, 2024, doi:10.3390/cells13211748_

Round 1

Reviewer 1 Report

Comments and Suggestions for Authors

Dear authors,

Congratulations on your excellent work!

I believe the manuscript would benefit from a brief description of the genetic abnormalities and alterations in signaling pathways and transcription factors in Hodgkin lymphoma.

Please consider including a discussion regarding telomeres structure, organization and regulation in normal cells.

Best regards!

Author Response

Reviewer 1:

I believe the manuscript would benefit from a brief description of the genetic abnormalities and alterations in signaling pathways and transcription factors in Hodgkin lymphoma.

We agree and have included these findings in the forelast paragraph of the Introduction (page 4, lines 9-20).

However, we feel that a too detailed description would be associated with the readers impression of “déjà vu” since we and others have dealt with these important facts in recent reviews.

Please consider including a discussion regarding telomeres structure, organization and regulation in normal cells.

We agree with the Reviewer and have added at the end of the Introduction (page 4) a new paragraph of 14 lines resuming the main components of the telosome (telomere/shelterin complex). Moreover, in the second paragraph of the heading 2.8. Prognostic and Future Potential Therapeutic implications of the 3D Nanotechnology Findings, (page 11, lines 15-19, new references 116-119) we discussed the most important finding, the high number of very short telomeres in H and RS cells of aggressive disease, in the setting of normal telomere biology.

Reviewer 2 Report

Comments and Suggestions for Authors

In the manuscript titled “3D-Q-FISH/Telomere/TRF2 Nanotechnology Identifies a Progressively Disturbed Telomere/Shelterin/Lamin AC Complex as the Common Pathogenic, Molecular/Spatial Denominator of Classical Hodgkin Lymphoma,” the authors present an overview of the application of 3D nanotechnology in assisting with the diagnosis of classical Hodgkin lymphoma. The article not only covers key pathogenesis steps (from LMP1 and telomere complex disruption to nuclear remodeling) but also introduces the diagnostic application by identifying the disruption of 3D telomere-TRF2 interaction. This comprehensive overview offers valuable insights for readers and is a publication-ready manuscript. However, there are some minor issues that need to be addressed before publication.

Reviewer comments:  

1.     The authors thoroughly discuss the significance of telomere alterations in the evolution of classical Hodgkin lymphoma (cHL). Through 3D reconstructive nanotechnology, they highlight a quantitative and qualitative disruption of telomere/shelterin interactions in tumor specimens. This article references key literature (Lab Invest. 2017 Jul;97(7):772-781; Transl Oncol. 2012;5:269-77) to demonstrate that a high percentage of telomere-stumps is a potential prognostic and possibly predictive factor in cHL patients. However, the quantification of telomere/shelterin disturbance relies on telomere distribution profiles, and no clear cut-off value is provided to predict patient prognosis at diagnosis. Additionally, this telomere signature appears complex and may not be practical for routine clinical use. The authors should explore ways to improve this limitation and discuss the optimal application of 3D nanotechnology for cancer patients. A section addressing these limitations and the future potential of this technology would strengthen the article.

2.     Certain careless errors were identified. Two short titles, “2.23. D Telomere-Shelterin complex: the railway turntable in H and RS morphogenesis “and “2.33. D nuclear remodeling during the transition from H to RS” should be “2.2. D Telomore….” and “2.3. D nuclear….” respectively.

3.     In 2.8 Prognostic and Future Potential Therapeutic implications of the 3D Nanotechnology Findings: “In a most recently published international, retrospective study [115] our pilot study findings [111] were confirmed.” Does author mean “ A recently published international retrospective study confirms the findings from our pilot study.” ?

Author Response

Reviewer 2:

  1. The authors thoroughly discuss the significance of telomere alterations in the evolution of classical Hodgkin lymphoma (cHL). Through 3D reconstructive nanotechnology, they highlight a quantitative and qualitative disruption of telomere/shelterin interactions in tumor specimens. This article references key literature (Lab Invest. 2017 Jul;97(7):772-781; Transl Oncol. 2012;5:269-77) to demonstrate that a high percentage of telomere-stumps is a potential prognostic and possibly predictive factor in cHL patients. However, the quantification of telomere/shelterin disturbance relies on telomere distribution profiles, and no clear cut-off value is provided to predict patient prognosis at diagnosis. Additionally, this telomere signature appears complex and may not be practical for routine clinical use. The authors should explore ways to improve this limitation and discuss the optimal application of 3D nanotechnology for cancer patients. A section addressing these limitations and the future potential of this technology would strengthen the article.

We appreciate very much the Reviewer’s comment and suggestion and have addressed this important issue in a new paragraph on page 11 (third paragraph, lines 23-39, references 120-123).

  1. Certain careless errors were identified. Two short titles, “2.23. D Telomere-Shelterin complex: the railway turntable in H and RS morphogenesis “and “2.33. D nuclear remodeling during the transition from H to RS” should be “2.2. D Telomore….” and “2.3. D nuclear….” respectively.

The corrections have been performed.

  1. In 2.8 Prognostic and Future Potential Therapeutic implications of the 3D Nanotechnology Findings: “In a most recently published international, retrospective study [115] our pilot study findings [111] were confirmed.” Does author mean “A recently published international retrospective study confirms the findings from our pilot study.”?

Yes, we agree. Since the sentence of the Reviewer is very clear and shorter we replaced our sentence with his sentence.